# Divergence Analyses of Sperm DNA Methylomes between Monozygotic Twin AI Bulls

**Shuli Liu [1], Siqian Chen [1], Wentao Cai [1] , Hongwei Yin [1], Aoxing Liu [1] , Yanhua Li [1,2], George E. Liu [3] , Yachun Wang [1], Ying Yu [1,* and Shengli Zhang [1,*]**

[1] Key Laboratory of Animal Genetics, Breeding and Reproduction, Ministry of Agriculture & National Engineering Laboratory for Animal Breeding, College of Animal Science and Technology, China Agricultural University, 2rd, Yuanmingyuan West Road, Beijing 100193, China; shuliliu1991@cau.edu.cn (S.L.); S20173040472@cau.edu.cn (S.C.); caiwentao@cau.edu.cn (W.C.); m18611323940@163.com (H.Y.); aoxing.liu@mbg.au.dk (A.L.); yhli1976@163.com (Y.L.); wangyachun@cau.edu.cn (Y.W.)

[2] Beijing Dairy Cattle Center, Qinghe South Town, Beijing 100085, China

[3] Animal Genomics and Improvement Laboratory, BARC, USDA-ARS, BARC-East, Beltsville, MD 20705, USA; george.liu@ars.usda.gov

\* Correspondence: yuying@cau.edu.cn (Y.Y.); zhangslcau@cau.edu.cn (S.Z.); Tel.: +86-10-62734611 (Y.Y.); +86-10-62733697 (S.Z.)

**Abstract:** Semen quality is critical for fertility. However, it is easily influenced by environmental factors and can induce subfertility in the next generations. Here, we aimed to assess the impacts of differentially methylated regions and genes on semen quality and offspring fertility. A specific pair of monozygotic (MZ) twin artificial insemination (AI) Holstein bulls with moderately different sperm qualities (Bull1 > Bull2) was used in the study, and each twin bull had produced ~6000 recorded daughters nationwide in China. Using whole genome bisulfite sequencing, we profiled the landscape of the twin bulls' sperm methylomes, and we observed markedly higher sperm methylation levels in Bull1 than in Bull2. Furthermore, we found 528 differentially methylated regions (DMR) between the MZ twin bulls, which spanned or overlapped with 309 differentially methylated genes (DMG). These DMG were particularly associated with embryo development, organ development, reproduction, and the nervous system. Several DMG were also shown to be differentially expressed in the sperm cells. Moreover, the significant differences in DNA methylation on gene *INSL3* between the MZ twin bulls were confirmed at three different age points. Our results provided new insights into the impacts of AI bull sperm methylomes on offspring fertility.

**Keywords:** monozygotic twins; Holstein bull; DNA methylome; differentially methylated regions; differentially methylated genes

## 1. Introduction

Decreased fertility or infertility is one of the most critical problems in human society and in the livestock industry. Twenty per cent of cases were fully attributed to the male factors, and another 30–40% were partially explained by male infertility [1]. In most cases, male infertility was coupled with decreased sperm counts and sperm motility. It is well established that aberrant epigenetics is related to male infertility, and one of the underlying mechanisms is environmental factors [2–4]. DNA methylation is one of the most stable epigenetic marks on the genome and can be a key regulator of transcription [5]. DNA methylation undergoes two waves of reprograming during mammalian development: one is in primordial germ cells, and another is after fertilization in the pre-implantation embryo [6]. Male germ cells are almost completely methylated at birth, except for some hypomethylated

regions escaping from reprograming. Of note, sperm methylome is a product of DNA methylation maintenance from birth. Environmentally induced DNA methylation changes could be accumulated during spermatogenesis given the continuous cycles of mitosis and meiosis [6]. Interestingly, epigenetic alteration in sperm could be transmitted to offspring and help to mediate the transgenerational inheritance [7]. For instance, Kropp et al. revealed that DNA methylation differences were associated with male fertility status and that embryonic transcriptional alterations might be influenced by the fertility status of the bulls [8]. Millissia et al. (2019) showed that exposure to toxicants (DDT) promotes the transgenerational inheritance of sperm DNA methylation differences, which originated during spermatogenesis in the testis [9]. In livestock, an elite dairy bull can have tens of thousands of daughters via AI, enabling robust observations of sperm qualities and daughters' performances. Extensive phenotype records, deep pedigrees and divergent selection enable cattle to be an ideal model for studying the association between sperm DNA methylation and male fertility.

Monozygotic (MZ) twins, who share nearly all of their genetic variants and similar early-life environments, provide a valuable resource for observation since many confounders like age, gender, and genetic background can be minimized [10]. MZ twins have been widely reported to be discordant in appearance, disease and other complex traits. Phenotypic discordance between MZ twins has been ascribed to the non-shared environmental exposure, in utero environment, de novo mutations and stochasticity [11,12]. We hypothesized that these factors may drive the epigenetic divergences between MZ twins and thus influence the performance of MZ twins in complex traits. For instance, in humans, Dempster et al. (2011) identified disease-associated methylation differences within MZ twins discordant for schizophrenia and bipolar disorder [13].

In the present study, we investigated the sperm DNA methylomes and transcriptomes of a pair of MZ AI twin bulls on genome-wide levels. Both the MZ twin bulls were AI sires in China, and each had around 6000 recorded daughters all over the country. We found that the MZ bull twins shared identical genetics and had very similar appearances, while they had differences for several important sperm quality traits and fertility traits between their daughters. We compared the DNA methylation and gene expression in sperm cells between the twin bulls and found that the differentially methylated regions (DMR) were closely associated with the expressions of spermatogenesis-related genes, such as *HSPA1L* and *ACTN1*. To understand whether the methylation differences would persist during a whole lifetime, we investigated the gene methylation at two other time points (57 months and 64 months of age) and observed that *INSL3* genes were consistently hypomethylated in the twin bulls. Collectively, our results demonstrated the importance of epigenetics on the performance of male fertility and highlighted several fertility-associated genes that were potentially epigenetically regulated.

## 2. Results

### 2.1. MZ Twin Bulls Were Genetically Identical but Discordant in Semen Quality

The MZ twin Holstein bulls (Bull1 and Bull2) from Beijing Dairy Cattle Center were used to analyze the sperm methylome. These twins, whose sire was a famous American AI sire (Blackstar, registration number HOUSA000001929410), arose from natural twinning after embryo transfer. Both of them had served in the Chinese dairy industry around ten years. As recorded by the Dairy Association of China, each of them had produced more than 300,000 straws of frozen semen throughout their life and had around 6000 daughters with nationwide distribution.

To confirm the genetic concordance of the twin bulls, genotypes of 54,609 single nucleotide polymorphisms (SNP) across the whole genome were analyzed using an Illumina Bovine SNP50 BeadChip. As a result, 51,370 loci were successfully genotyped in both bulls and, of those, 99.99% (51,364 loci) were identical. Although the remaining six loci showed different genotypes within the BeadChip, they were subsequently confirmed to be identical by Sanger sequencing (Figure 1A). Thus, the two bulls were confirmed to be MZ twins.

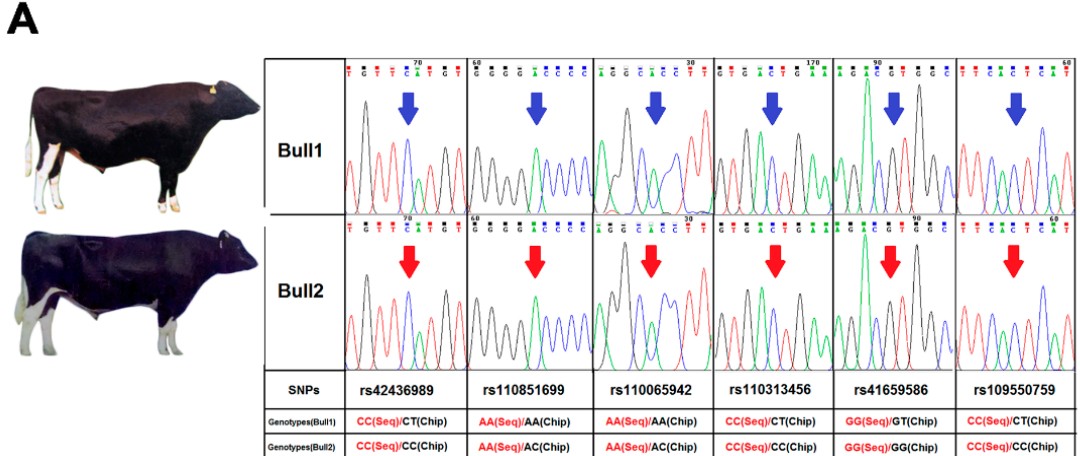

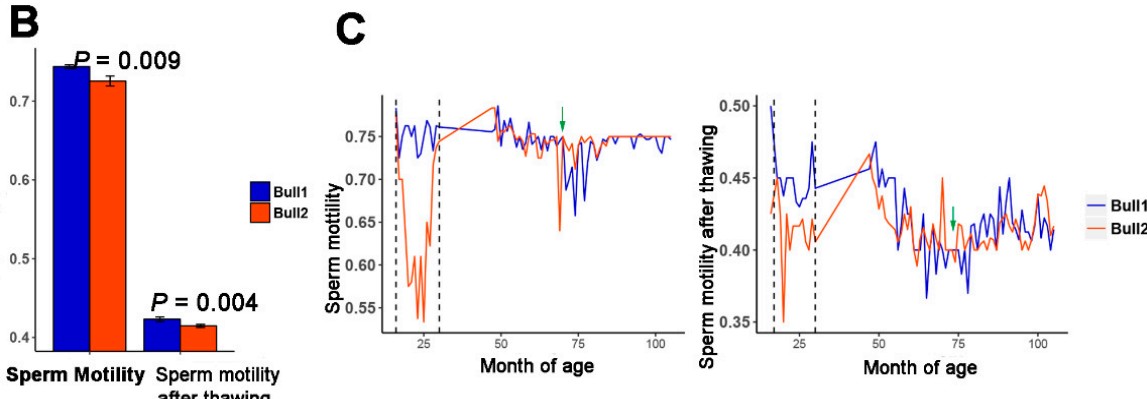

**Figure 1.** Comparison of sperm traits of the twin bulls. (**A**) Six SNPs were verified as identical through Sanger sequencing. The blue and red arrows indicate the SNP loci. "Seq" indicates detected genotypes using Sanger sequencing, and "Chip" indicates detected genotypes using the Illumina Bovine SNP50 BeadChip. (**B**) Two traits (fresh sperm motility and sperm motility after thawing) were compared between twin bulls using a paired *t*-test based on monthly average (data for 16 to 105 months of age were available except for 31 to 46 months of age). Both these two traits showed significant differences (*p* = 0.009, 0.004). (**C**) Sperm motility changes throughout their working months. The green arrows indicate the month of sample collection.

To compare the phenotypic differences between the twin bulls, we firstly collected four types of semen quality traits along their whole lives. We observed that the average levels of fresh sperm motility and sperm motility after thawing of Bull1 were significantly superior to those of Bull2 (*p* < 0.01) (Figure 1B). Only these two sperm traits in the first 15 months of semen collection (16–30 months of age) were significant between them (Figure 1C). Other semen quality traits between the twin bulls did not show significant differences. We further compared the estimated breeding values (EBVs) of eleven female fertility traits between the two twins based on their daughters' performance (1064 and 1279 daughters for Bull1 and Bull2, respectively) [14], including five traits from heifers and six traits from cows. The reliabilities of the female fertility EBVs ranged from 0.67 to 0.99. The results showed that for most female fertility traits (10/11, 90.9%), Bull1 performed better than Bull2 (Figure 2, Table S1). Therefore, the MZ twin bulls were concordant in genotypes but discordant in semen quality traits.

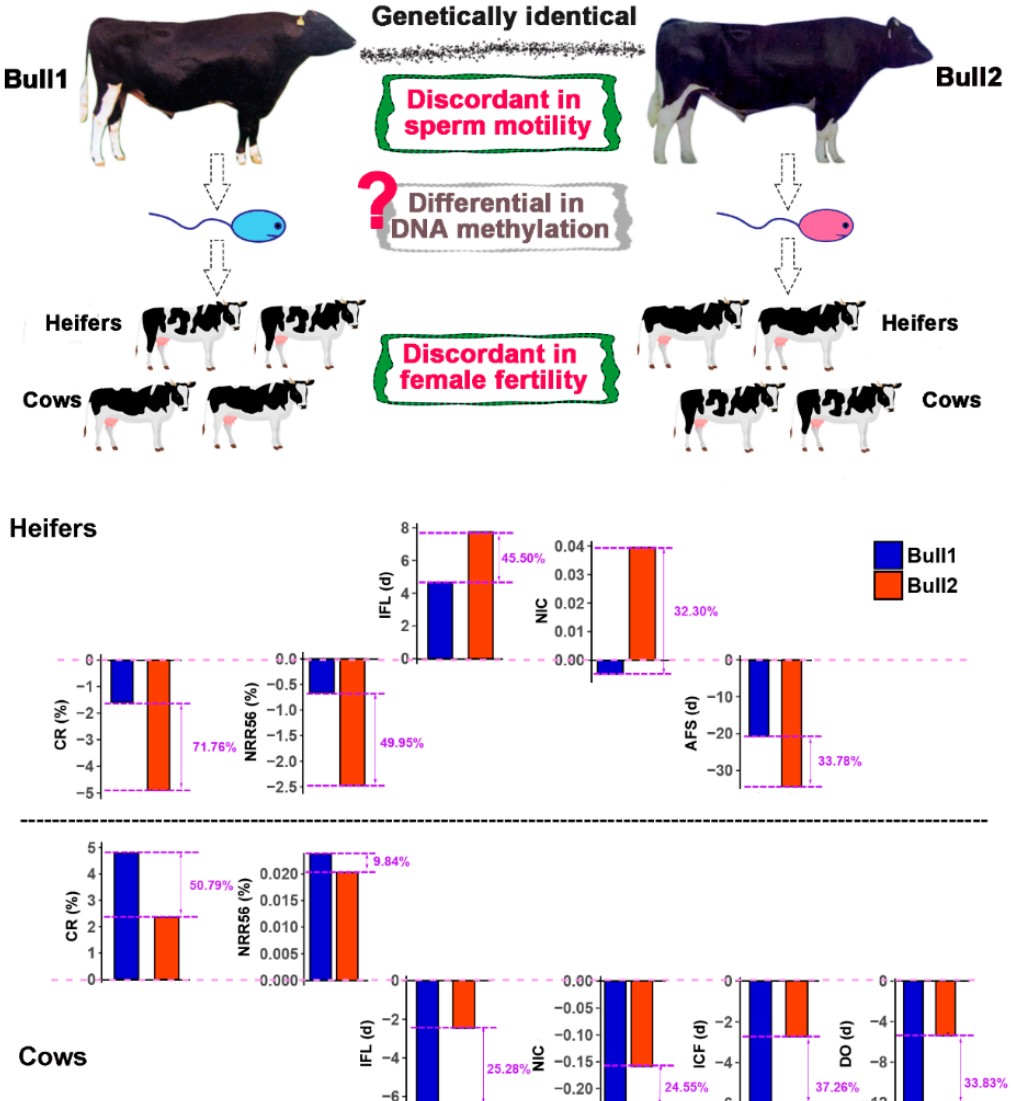

**Figure 2.** Estimated breeding values (EBVs) of female reproductive traits of the twin bulls based on their daughters. EBVs of eleven female fertility traits (five heifer and seven cow traits) were estimated by the DMU package (Version 6, release 5.2) [15], using data from 153,819 daughters derived from families of 4302 bulls. EBVs are shown as the levels above or below the population average (pink lines). The numbers in purple color are the percentages of the deviation of EBVs between the twin bulls normalized by the genetic standard deviations. Except AFS for heifers, the performances of Bull1 were better than Bull2 in all other ten traits. AFS: age (day) at first insemination; IFL: interval from first to last insemination; CR: conception rate for first insemination; NIC: number of inseminations per conception; NRR56: non-return rates within 56 days after first insemination; ICF: interval from calving to first insemination (days); DO: days open, which indicated dry period. Detailed information can be found in Table S1.

### 2.2. Landscapes of the Sperm DNA Methylomes

We investigated the sperm DNA methylomes of the MZ twin bulls using whole-genome bisulfite sequencing (WGBS) at single-base resolution. We generated 526 and 582 million unique mapped reads and achieved 23.5× and 26× average read depths for Bull1 and Bull2, respectively. The sequencing details are summarized in Table S2. Significant methylation differences at non-CpG sites (CHG, CHH, H = A/C/T) were not observed for the twin bulls, thus we only included genome-wide CpG sites in the subsequent methylation analyses.

In total, we determined the methylation status of about 92.5% of bovine genomic CpG sites (~27 million CpG sites, with sequence coverage of at least five reads) in the sperm samples of the twin bulls. Of these, 72–82% of all analyzed CpG sites were either extremely hypomethylated (~10%, methylation level <5%) or strongly hypermethylated (~70%, methylation level ≥80%) (Figure 3A). A global-scale view revealed that CpG methylation levels exhibited large variations throughout each chromosome (Figure S1). To characterize the dynamics of methylation in the bovine sperm genome, we determined the average methylation levels in gene bodies and the regions 2 kb upstream and downstream of these gene bodies. We observed decreased CG methylation levels in the gene promoters, which reached their nadir at the transcriptional start site (TSS), followed by consistently increasing methylation levels from the TSS until the transcriptional termination site (TTS) (Figure 3B). These observations were consistent with those in other species, indicating a conserved methylation pattern along the genic regions [16–18]. We further investigated the promoter methylation levels of different gene categories. A promoter region was defined as 2 kb upstream of the TSS. Protein-encoding genes were less methylated and exhibited high variation in their methylation level (Figure 3C). In contrast, snoRNAs, snRNAs, tRNAs, and rRNAs, which tend to be constitutively expressed and independent of DNA methylation regulation, were highly methylated (Figure 3C) [19]. Hypermethylation of miRNAs (the first boxplot in Figure 3C) and pseudogenes (the fourth boxplot in Figure 3C) may suppress the expression of these genes, which are important to maintain the stability of the bovine sperm genomes [18].

To confirm the results of WGBS, we conducted targeted bisulfite pyrosequencing on four randomly chosen regions, including 20 CG sites. The correlation between the methylation levels from WGBS and bisulfite pyrosequencing was high, with a Pearson correlation coefficient of 0.966 ($p = 7.52 \times 10^{-24}$) (Figure 3D), confirming the accuracy of the genome-wide DNA methylation data in this study.

### 2.3. DNA Methylation Conservation and Divergences between the MZ Twin Bulls

To unravel the DNA methylation conservation patterns among individuals, we correlated the DNA methylation levels between the twin bulls, in different chromosomes, functional elements and repeats. Our results showed that the sperm DNA methylation of the twin bulls was highly correlated with each other on a genome-wide level (Pearson's $r = 0.90$) (Figure 4A), implying that our datasets had accurate callings (Figure 4A). We observed that promoters and the 5′-UTR showed the highest correlation coefficients (Pearson's $r = 0.97$), indicating the conservation of DNA methylation at gene-regulatory elements (Figure 4A) [20]. Repeat elements were mostly dynamic with the lowest correlation coefficients (Pearson's $r = 0.87$). Of interest, we observed that DNA methylation levels of CpG islands (CGI), TSSs and 5′ exon–intron junctions were the highest correlated between the individuals, which were compatible with those between different human cells (H1 human embryonic stem cells and IMR90 fetal lung fibroblasts) [16] (Figure 4B).

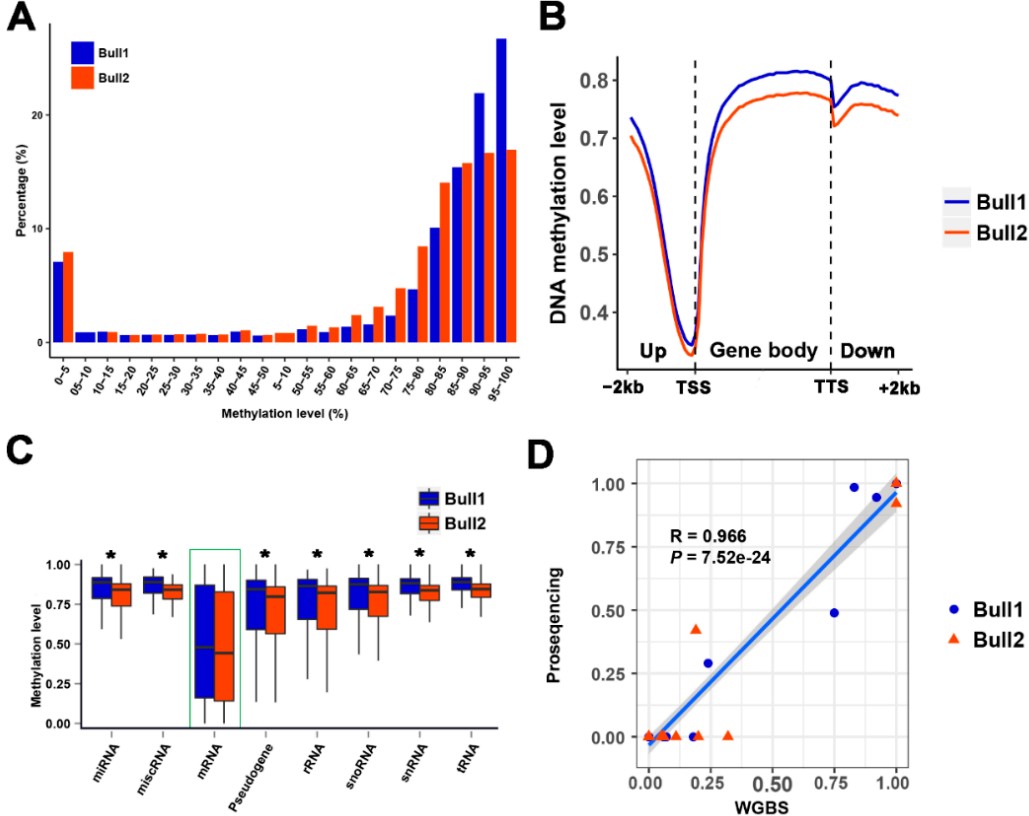

**Figure 3.** DNA methylation profiles along the bovine genome. (**A**) Distribution of methylation levels of CpG sites (only CpG sites covered by at least five reads were included). (**B**) Average methylation levels in promoters (−2 kb), gene bodies and downstream (2 kb) regions in sperm samples of the MZ twin bulls. We divided the promoters and downstream regions of genes into 20 equally sized bins, and the gene bodies into 40 equally sized bins. Average methylation levels (the ratio of methylcytosines to totally sequenced cytosines) of corresponding bins were calculated. (**C**) DNA methylation levels of different gene categories in promoters. Box plots show the methylation level of each gene category. Each category was compared with coding protein (mRNA). The asterisks (*) mean that the promoter methylation differences between mRNA and other gene categories were significant ($p < 2.2 \times 10^{-16}$). (**D**) Pearson correlation coefficient of methylation levels of 20 CpG sites derived from WGBS and pyrosequencing. Each dot represents one or more CpG sites (there are some dots that are overlapped).

To investigate the divergences of DNA methylation, we detected 522,722 differentially methylated cytosines (DMCs; referring to cytosines in CG content) (coverage ≥5; Fisher's exact test, FDR < 0.05) between the twin bulls, with 88.17% hypermethylated and 11.83% hypomethylated in Bull1. The DMCs were spread widely across the genome, with enrichment in intergenic regions and repeats but depletion in CpG islands, promoters, and exons ($p < 1 \times 10^{-50}$, Fisher's exact test in R) (Figure 4C), as well as in BTA18, 19, 25, and the X chromosomes (*$p < 1 \times 10^{-50}$, Fisher's exact test in R) (Figure 4D).

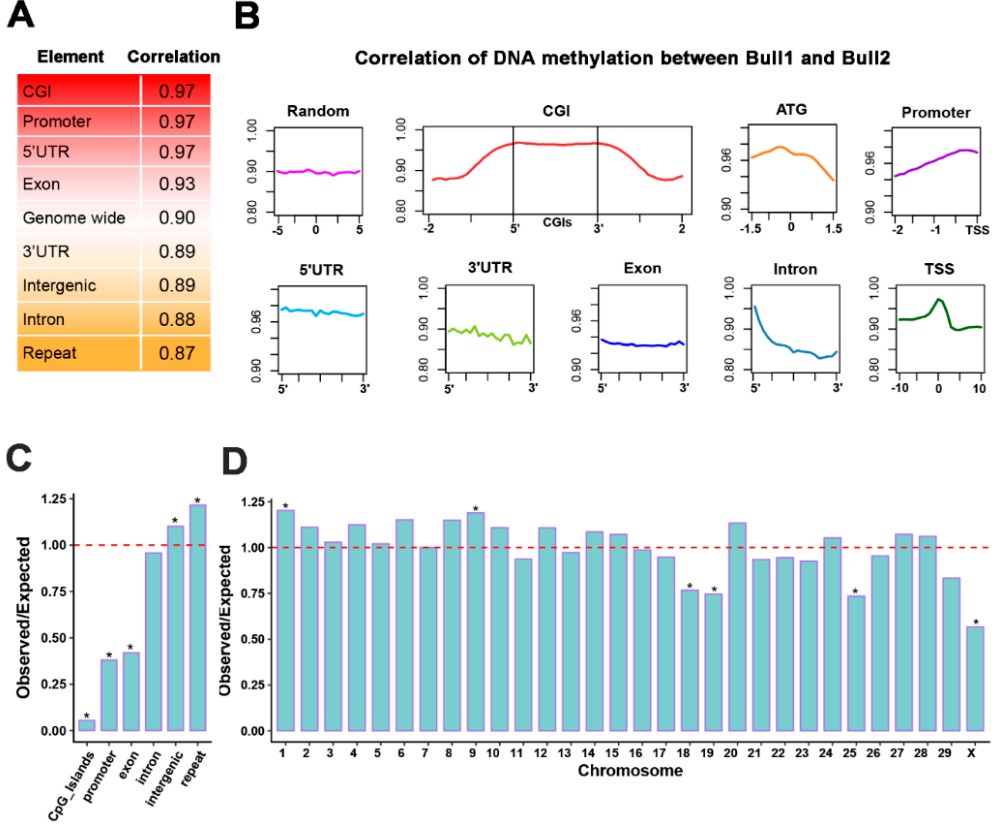

**Figure 4.** DNA methylation conservation and divergences between the sperm genomes of the MZ twin bulls. (**A**) Pearson correlation coefficients in nine genome elements were calculated based on CpG sites with at least five read coverages. (**B**) Dynamic changes of Pearson correlation coefficients along each genome element. Each element was divided into 20 equally sized bins. Correlation rates were calculated in each bin using CpG sites with at least five read coverages. (**C**) Enrichment analysis of DMCs in six genome elements. Expected DMCs at a random distribution were computed based on the number of CpG sites in each element and total number of DMCs. Observed DMCs were compared to expected DMCs, and *p* values were calculated (*$p < 1 \times 10^{-50}$, Fisher's exact test in R). (**D**) Enrichment analysis of DMCs in 29 autosomes and X chromosome (*$p < 1 \times 10^{-50}$, Fisher's exact test in R).

## 2.4. Differentially Methylated Regions (DMR) between the Twin Bulls Were Associated with Reproduction and Nervous Development

To dissect the discordant regions of sperm DNA methylation associated with male fertility, we identified DMR between them using the MethPipe [21] at a stringent threshold of >0.3 for mean CpG methylation differences in a DMR region. We identified 528 DMR between the twin bulls, of which 369 were hypermethylated and 159 were hypomethylated in Bull1 compared with Bull2 (Figure 5A, Table S3). Of the total DMR, 55.7% were overlapped with one or more genes in promoters (2 kb upstream to TSS) or genes bodies (Table S4) corresponding to 309 differentially methylated genes (DMG). Notably, functional enrichment analysis revealed that the DMG with DMR in the promoters and 5′-UTRs (*n* = 85) were particularly associated with gene ontology (GO) terms relevant to sex determination and fertility, such as sex differentiation, gonad development, Leydig cell differentiation and reproductive processes (FDR < 0.1; Figure 5C, Table S5). In addition, transcription factor binding analysis in DMR-overlapped promoter sequences revealed the enrichment of putative binding sites of a transcriptional factor (*EGR1*), which is known for its key function in cell proliferation and differentiation [22]. Moreover, genes with a DMR in an exon and the 3′-UTR were enriched in GO terms of cell differentiation and development, transcription, embryo development, reproductive system development, animal organ development as

well as nervous system development (FDR < 0.1; Figure 5D, Table S5). However, genes with a DMR in an intron region failed to yield significant enrichments (Table S5).

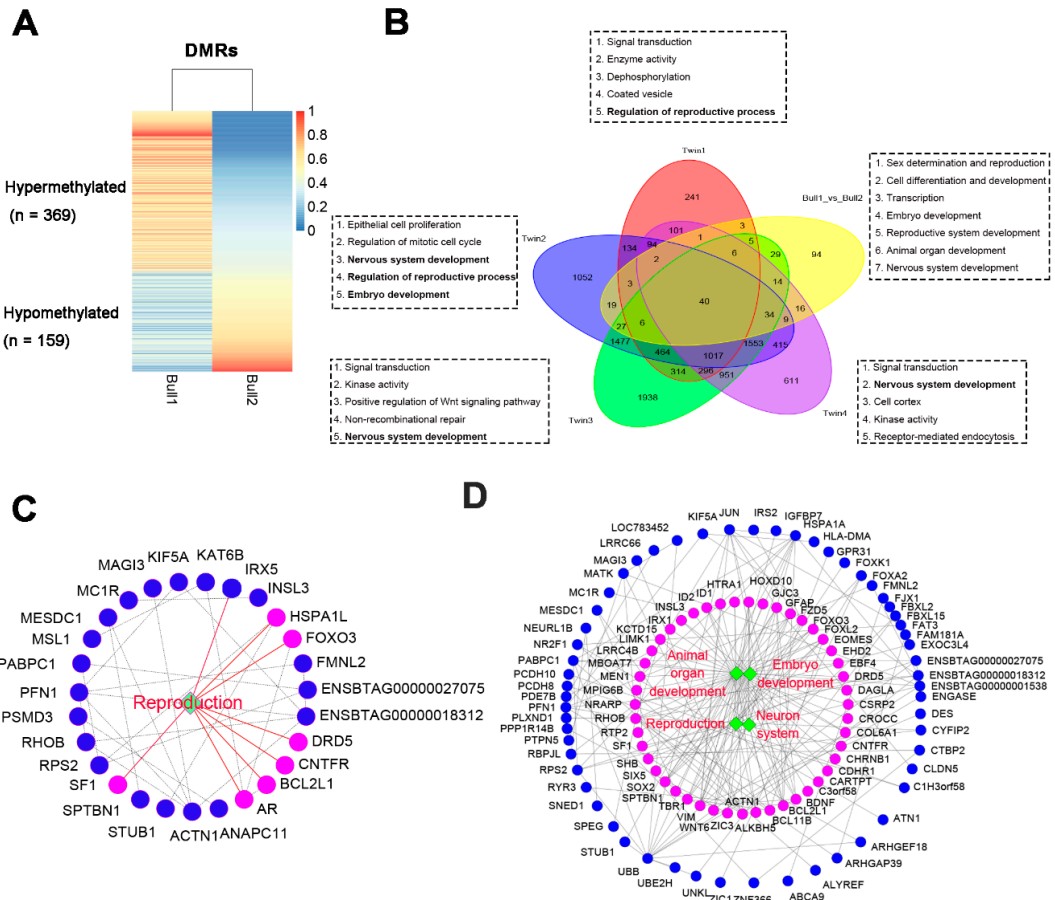

**Figure 5.** Differentially methylated regions (DMR) and differentially methylated genes (DMG) between the twin bulls. (**A**) The heat map of the methylation levels of DMR of the twin bulls. (**B**) The overlapped DMG and enriched functions among five pairs of MZ twin bulls. The methylation data of the other four pairs were downloaded from GSE74200. Enriched functions were shown for each pair of MZ twin bulls. Functions with bold fonts were also discovered in our study. (**C**) DMG with DMR in promoters and 5'-UTRs, enriched in functions related to reproduction. (**D**) DMG with DMR in exons and 3'-UTRs, enriched in functions related to animal organ development, reproduction, embryo development and the neuron system (right). Solid circles indicate genes, while the pink ones represent genes enriched in specific functions.

It is noted that due to the limited sample size (only one twin pair), methylation differences could be induced by stochastic effects. To validate our results, we compared the DMG in our study with those between four other pairs of MZ twin bulls divergent in sperm quality from Shojaei Saadi et al. (2017) [23]. There was one agreement in DMG-enriched GO categories between the four twin pairs from the study of Shojaei Saadi et al. (2017) and the twin bulls in our study, namely, the reproduction, embryo development and neuron systems (Figure 5B). By intersecting all DMG of all five twin pairs, we identified 40 DMG in common. These sperm and fertility-related DMG in bovines are listed in Table S6.

To validate the methylation differences of DMR, we randomly chose two DMR regions located in the promoters of *INSL3* and *PSMD3* genes to conduct targeted bisulfite cloning sequencing. The experimental results further confirmed that the two DMR located in the gene promoters were indeed differentially methylated (Figure 6A,B).

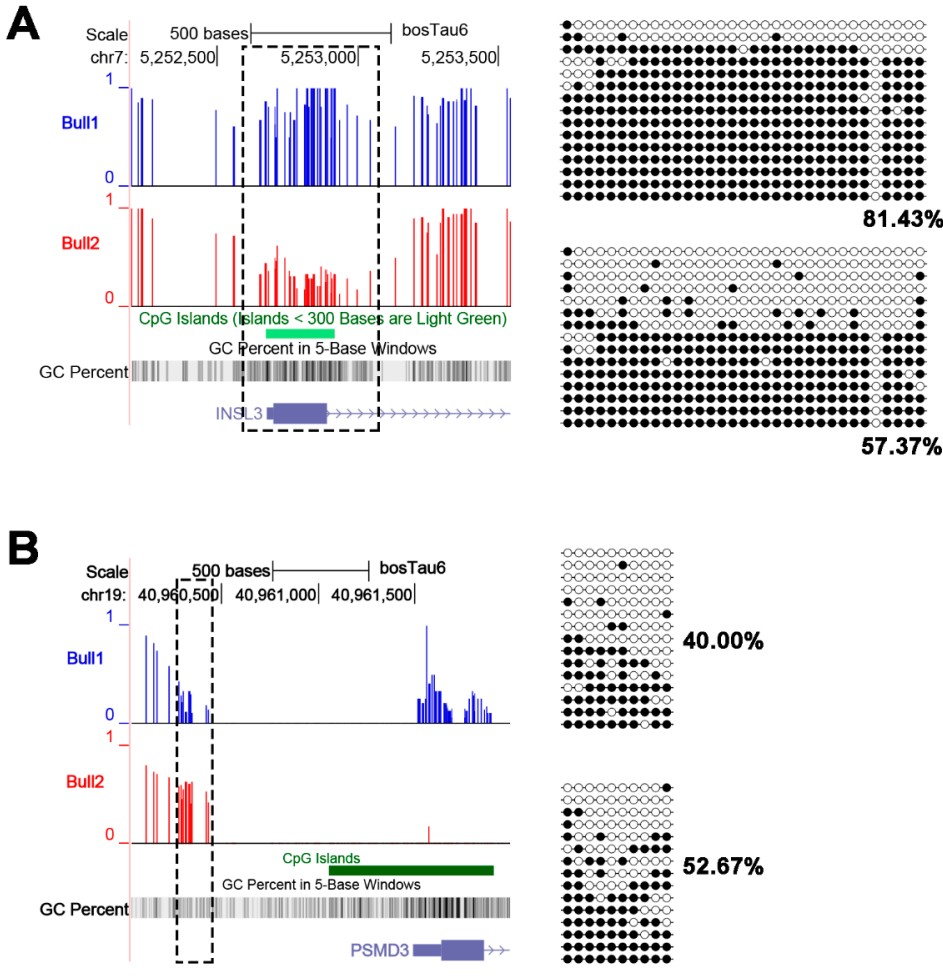

**Figure 6.** Two randomly chosen DMR located in gene promoters were validated by bisulfite sequencing. Figures to the left show the methylation levels from WGBS, and figures to the right show the methylation levels of DMR from bisulfite sequencing. Primers are listed in Table S1.

*2.5. Inverse Correlation between Promoter Methylation and Gene Expression in Sperm Cells*

To explore the relationship between DNA methylation and gene expression, the transcriptome data for the twin bulls were generated using RNA-seq. We divided all genes ($n$ = 24,616) equally into five categories (lowest, lower, medium, higher and highest) based on their expression levels and then explored the distribution of the methylation level from 4 kb upstream to 2 kb downstream of the gene. We observed a significant negative correlation in gene promoters (from −1.5 kb to TSS) (Figure S2A,B) and a positive correlation in the genes body regions, consistent with previous studies in other species (Figure S2A,C). Furthermore, the region from −300 to 200 bp of TSS achieved the strongest negative correlation (Pearson's correlation $r$ = −0.1172, $p < 2.2 \times 10^{-16}$). The above results indicated that the DNA methylation levels of bovine sperm cells could partially explain the divergences of gene expression patterns. We identified 1193 (detected by Cuffdiff, $p$-value < 0.05, |log$_2$FC| > 2), 4692 (detected by DEGseq, $q$-value < 0.001, |log$_2$FC| > 2), and 1997 (detected by edgeR, $q$-value < 0.05, |log$_2$FC| > 2) DEG between the twin bulls using the three analysis approaches. Among these, 44 DEG that were simultaneously identified by two or three approaches were also differentially methylated. Among them, several DEGs (*HSPA1L*, *ACTN1*, *PSMD3*, and *CSRP2*) were negatively associated with DNA methylation of their promoter regions (Figure S3A–D). In particular, *HSPA1L* (heat shock 70 kDa protein 1-like) and *ACTN1* (actinin alpha 1) have been suggested to be associated with spermatogenesis [24,25]. *HSPA1L* was further confirmed to be more highly expressed in spermatozoa of Bull1 than in Bull2 via qRT-PCR, and was also shown to exhibit specifically higher expression in spermatozoa than in

other tissues (Figure S3E). These findings suggest that *HSPA1L* may play a key role in the fertility discordance between the MZ twin bulls.

*2.6. Persistence of Sperm DNA Methylation Differences across Different Ages*

To test the hypothesis that some methylation differences could observed during a whole lifetime, we chose two more age points (57 months and 64 months of age) for the MZ twin bulls to investigate the sperm methylation level of a validated DMR region. The DMR was located at the promoter regions of *INSL3*. Consequently, we found that the DMR region located in the promoter of *INSL3* was lower methylated in Bull1 than in Bull2 at the two ages (Figure 7). These observations were in concordance with the methylation alterations at 72 months of age (Figure 6A), implying that there was a stable divergence of DNA methylations between the MZ twin bulls at different ages.

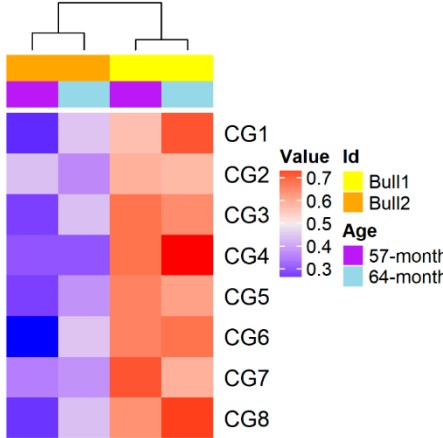

**Figure 7.** Methylation differences of a DMR observed at another two time points (57 and 64 months of age).

## 3. Discussion

The current study combined the sperm DNA methylome and transcriptome in a pair of MZ twin AI bulls to investigate the relationship between methylation variations and semen quality traits as well as daughters' fertility traits. We identified 528 DMR among MZ twin bulls, which were overrepresented in gene bodies/gene promoters. We observed that 309 DMG were highly enriched in several important GO terms, including reproduction. Among them, we revealed that *HSPA1L*, *ACTN1* and *INSL3*, which play critical roles in spermotogenesis, might be candidate genes for epigenetic regulation of male fertility. Our results provided insights into the DNA methylation differences associated with male fertility using the strategy of MZ twin bulls.

Male fertility is a complex trait. Determining the underlying causes of suboptimal reproductive performance remains a challenge in the dairy industry and the human reproductive medicine system. In the dairy cattle production system, with the wide-spread adaptation of the AI industry, a side effect coupled with the improved dairy production efficiency is the negative impact on reproduction performance [26]. In addition, the extensive selection of milk production in dairy cattle also deteriorates the fertilizing capacity [27]. Multiple measurements for male fertility are used, such as semen quality and counts in each ejaculate, sperm morphology, sire conception rate (SCR) and scrotal circumference. The low heritability of the above traits limits the improvement of male fertility via genetic improvement, even though large effects have been observed [28–30]. Increasing evidence has pointed towards casual effects of epigenetic regulation on male fertility. In human studies, DNA methylation is one of the most studied epigenetic modifications. Urdinguio et al. (2015) identified aberrant DNA methylation of Alu Yb8 repeats in infertile men [31]. Tang et al. (2018) revealed DMR in three imprinted genes, *H19*, *GNAS* and *DIRAS3*, between fertile controls and infertile patients [32]. Sujit et al. (2018) associated several imprinted genes, spermatogenesis-associated genes and immune response genes with male

infertility [33]. DNA methylation changes associated with male fertility could also be accumulated along with increasing age [34]. A longitudinal study revealed that MZ twins show more differences in DNA methylation as they get older [34]. Therefore, the DNA methylation differences identified in our twin samples were a collection of methylation changes during their individual developments.

DNA methylation undergoes two waves of epigenetic reprogramming: One is genome-wide demethylation during primary germ cell development and remethylation from the prospermatogonia stage [35]. Another is postzygotic demethylation and remethylation [35]. Mature germ cells can be considered as the terminal products of these processes in mammals [36]. Although highly methylated, sperm DNA still has regions that are protected and hypomethylated. Some promoter regions in sperm were marked by hypomethylation and retained nucleosomes, which have been proposed to aid rapid activation during early embryo development after fertilization [36]. Aberrant sperm DNA methylation, which might be induced by environmental factors, like diet, physical activity, exposures to toxic substances and others, would affect male fertility status as well as embryo quality [8,37,38]. Environmental-driven epigenetic changes in sperm may even resist reprogramming and persist into successive generations' offspring, through mechanisms which are likely independent from genetic factors and may help to partially explain the "missing heritability". For instance, aberrant sperm DNA methylation of imprinted genes (*H19*, *Meg3*, *Peg1* and *Peg3*) could pass through two generations in a mice model [39]. Besides imprinted genes, transgenerational sperm DNA methylation was also observed in non-imprinted genes associated with spermatogenesis in the testis, following ancestral vinclozolin exposure [38]. In our study, by minimizing the genetic effect, we observed discordances in female fertility traits of daughters between twin bulls, which might be related to transgenerational epigenetic inheritance. Further works are needed to explore DNA methylation differences in the tissue samples of these female offspring.

Our results indicated that the DMR were particularly associated with genes relevant for reproduction, animal organ development, embryo development, and the nervous system, which is consistent with previous studies. Verma et al. discovered 151 genes differentially methylated in the sperm of highly fertile and subfertile buffalo bulls using a custom-designed microarray, of which 13 have roles in sperm functions and embryogenesis [40]. In addition, by comparing the sperm methylome of fertile and infertile humans using the Illumina 450 k array, Camprubi et al. revealed 696 differentially methylated CpG sites associated with 501 genes, for which gene ontology enrichment analysis revealed their associations with processes related to spermatogenesis [41]. Recently, Kropp et al. also assessed the DNA methylation of spermatozoa between high- and low-fertility bulls, revealing 76 DMR associated with 23 genes [8]. Fang et al. (2019) revealed DNA methylation differences associated with sire conception rate (SCR) enriched in signals of genome-wide association studies for reproduction traits [42]. Low SCR-specific partially methylated regions (PMDs) showed enrichment in GO terms associated with embryo system development [42]. Upon comparison, the 14 DMG revealed in the above four studies were also discovered to be differentially methylated in our analysis, namely, *BCL2L1*, *ID1*, *PADI2*, *ATN1*, *CELF6*, *DOCK2*, *EHD2*, *FOXK1*, *GSG1L*, *HSPA1L*, *HSPA1A*, *PDE7B*, *THRB*, and *ZFYVE28*, revealing their potential role in the regulation of male fertility.

We revealed several potential candidate genes, such as *HSPA1L*, *ACTN1* and *INSL3*. The DMR corresponding to *HSPA1L* gene was within a CpG island and located about −800 kb upstream of the TSS for *HSPA1L* and in exon 1 for *HSPA1A*. In agreement with our results, Camprubi et al. revealed CpG sites within exon 1 of *HSPA1A* and in the promoter of *HSPA1L*, which were differentially methylated between fertile and infertile men [41]. The promoter region of *HSPA1L* in the mouse genome has also been shown to exhibit testis-specific methylation status (unmethylated in the testis and methylated in other tissues), while *HSPA1L* was reported to be exclusively expressed in the testis [43,44]. *HSPA1L* is mainly expressed in the postmeiotic phase during spermatogenesis and may be involved in the binding of spermatozoa to the zona pellucida [45]. Its absence leads to male infertility [46]. We observed lower methylation in the promoter and high expression of *HSPA1L* in the individual with better performance in reproductive traits (i.e., Bull1), which was consistent with our expectations. *ACTN1* showed the

opposite trend of being highly methylated in the promoter and expressed at a lower level in Bull1, which coincided with the better performance in reproductive traits. *ACTN1* has also been proven to play pivotal roles in sperm capacitation and acrosome reaction [47,48]. The protein encoded by *ACTN1* is a member of a focal adhesion complex present in the spermatozoa. This complex takes part in remodeling of the actin cytoskeleton in mammalian spermatozoa, which is essential for the acrosome reaction and for sperm to achieve adequate motility [47,48]. It was also reported that polymorphisms of *ACTN1* were associated with fertility traits (non-return rates and number of piglets born alive for boars) in an association study [49]. However, the mechanisms by which *ACTN1* methylation regulates reproductive ability in dairy cattle require further analysis. Another interesting gene, *INSL3*, secreted uniquely by mature Leydig cells, plays roles in testicular descent during fetal development [50,51] and follicle development during the estrous cycle in female mammals [52]. These genes are of great importance to the epigenetic regulation of reproduction, and they require further functional validation.

We compared our results with those of Shojaei Saadi et al. (2017), who investigated the methylation differences within four pairs of male fertility-discordant MZ twin bulls. We observed 40 DMG that overlapped among the five pairs of MZ twin bulls. However, most of the DMG (259/309) identified in our studies were not differentially methylated in the other four pairs of male fertility-discordant MZ twin bulls [23]. This could be attributed to the different strategies in DNA methylation analysis. Shojaei Saadi et al. (2017) limited their analysis to the targeted regions of a custom-designed microarray, while our study conducted a whole genome-wide analysis using WGBS. In addition, the ages at the time of semen collection were different for three of the four MZ twin pairs in the study of Shojaei Saadi et al. (2017) [23], which could be another reason for the different DMG detected between the two studies since DNA methylation changes with age. In contrast, the semen samples were collected within one week in our study. By comparison, the limitation of our study was that only one pair of MZ twin bulls was investigated. DNA methylation differences could be driven by the stochastic factor. A larger sample size is required to further investigate the DNA methylation differences associated with bull fertility.

## 4. Materials and Methods

The following protocols were approved by the Institutional Animal Care and Use Committee (IACUC) of China Agricultural University (Permit Number DK996,30 September 2006). All the experiments were performed in strict accordance with the regulations and guidelines established by this committee.

### 4.1. Collection of Semen Samples and Semen Fertility Records of the MZ Twin Seed Bulls

Semen samples of the two Holstein MZ twin bulls were obtained from Beijing Seed Bull Station, China. The fresh semen samples were diluted in a solution containing milk, yolk, fructose, sodium citrate, and glycerin, among others, and then cryopreserved as granules. Each granule had about $1 \times 10^7$ sperm cells. The frozen semen samples at 72 months of age were collected on two days within the same week for the WGBS assay. In addition, the frozen semen samples at 57 and 64 months of age were also collected for DNA methylation verification.

Semen fertility records of the MZ twin seed bulls, including fresh sperm motility, sperm motility after thawing, sperm concentration and sperm volume per ejaculate, were collected from 16 months of age, when the twin bulls began semen collection, to 72 months of age. In total, we collected 711 and 887 records for the twin bulls in relation to each semen fertility trait.

### 4.2. Sperm DNA Extraction and Library Generation for WGBS

Sperm DNA was extracted using a salt fractionation protocol. A total of 5.2 µg of genomic DNA spiked with 26 ng of lambda DNA was fragmented into 200–300 bp, followed by end repair and adenylation. Cytosine-methylated adapters provided by Illumina were ligated to the sonicated DNA. Then, these DNA fragments were treated twice with bisulfite using an EZ DNA Methylation-Gold™

kit (Zymo Research, Irvine, CA, USA), as per the manufacturer's instructions. The resulting DNA fragments were amplified by two cycles of PCR using KAPA HiFi HotStart Uracil + ReadyMix. The libraries were sequenced on an Illumina Hiseq 2500 platform and 125 bp paired-end reads were generated (Novogene Inc., Beijing, China).

### 4.3. Data Analysis of WGBS

Raw data in FastQ format were first preprocessed to remove low quality reads. After filtering, the clean reads were aligned to a reference genome (UMD3.1) with the default parameters using Bismark bowtie2 [53,54] (version 0.14.5). Deduplicated reads were used to calculate sequencing depth and coverage of CG sites. The sodium bisulfite non-conversion rate was calculated as the percentage of cytosines sequenced at cytosine reference positions in the lambda genome.

Hypomethylated regions (HMRs) and differentially methylated regions (DMR) were identified using the software MethPipe (version 3.4.2) [36]. To obtain high confidence, we set a stringent threshold of > 0.3 for mean CpG methylation difference and a minimum of five differentially methylated cytosines (DMCs) in a DMR region. Genes overlapping with DMR in promoters (−2000 bp to TSS) or gene bodies were recognized as differentially methylated genes. Gene ontology (GO) analyses were conducted using KOBAS 3.0 [55]. We also predicted transcription factor enrichment by Regulatory Sequence Analysis Tools (RSAT, 2016) [56]. In brief, promoter sequences overlapping with DMR were submitted to a peak-motif ChIP-seq analysis pipeline. Then, the discovered motifs were compared with the JASPAR core vertebrate database (http://jaspar.genereg.net/cgi-bin/jaspar_db.pl). We identified significant transcription factors (TF) at a Pearson correlation threshold of 0.8.

### 4.4. WGBS Validation Using Pyrosequencing, Bisulfite Sequencing and Sequenom MassArray

Quantification of DNA methylation was carried out by pyrosequencing, as previously described [57]. Pyrosequencing primers were designed using Qiagen PyroMark™ Assay Design software 2.0. Primer sequences are given in Table S7. The biotinylated strand was purified using streptavidin Sepharose high-performance beads (GE Healthcare) and a PyroMark™ MD pyrosequencer (Biotage, Charlotte, NC, USA) using PyroMark™ Gold Q96 SQA reagents (Qiagen, Shanghai, China), with the quantification of methylated and unmethylated alleles using Pyro Q-CpG 1.0.9 software (Biotage).

For bisulfite sequencing, primers were designed by MethPrimer 2.0 [58]. Primer sequences are given in Table S7. PCR products were purified using Nucleospin Gel and PCR clean up kit (Macherey-Nagel GmbH, Düren, Germany), subcloned into PMD-18T vector, and transferred into Trans1-T1 Phase Resistant competent cells. Plasmid DNA containing PCR fragments was isolated from positively transformed colonies and sequenced.

The primers used in Sequenom MassArray were designed with an online software (EpiDesigner, http://www.epidesigner.com/), and the sequences of the primers are listed in Table S7. PCR products were purified using Shrimp Alkaline Phosphatase (SAP) and digested by RNase A. The DNA methylation of fragmented samples was detected using Mateix-Assisted Laser Desorption/ Ionization Time of Flight Mass Spectrometry (MALDI-LOF MS, Compass Inc., Beijing, China) and the data were visualized by EpiTYPER analyzer software v.1.0.5 (Sequenom, San Diego, CA, USA).

### 4.5. Sperm RNA Sequencing

Total RNA was extracted from sperm cells using the standard TRIzol method for three replicates per sample. Since there is only 30–200 fg of total RNA per bovine sperm cell, which is much less than in somatic cells, we first preamplified extracted RNA using the Smart-seq2 protocol following the manufacturer's instructions [59–62]. Because Smart-seq2 is limited to poly(A)$^+$ RNAs, we aimed to discover mRNAs and lncRNAs containing poly(A)$^+$. A total of 20 ng of cDNA was subjected to library construction based on the manufacturer's instructions. Sequencing was conducted in Hiseq2500 and 250 bp paired end reads were obtained.

### 4.6. Sperm RNA-Seq Analysis

The raw sequences produced by the Illumina pipeline in FastQ format were first preprocessed, as described in the WGBS analysis, to ensure the quality of data used in further analysis (Annoroad Inc., Beijing, China). After filtering, the clean reads were mapped to the reference genome (UMD3.1) using TopHat v2.0.12 (www.ccb.jhu.edu/software/tophat/index.shtml) [63]. Read counts for each gene in each sample were obtained using HTSeq v0.6.0 (https://htseq.readthedocs.io), and FPKM (fragments per kilobase of transcript per million mapped reads) was then calculated to estimate the expression level of genes in each sample. After checking the clustering of the three replicates in each sample, we merged the bam files of each sample into one file. Read counts for each gene in each sample were then counted by HTSeq v0.6.0 for subsequent analyses of differentially expressed genes (DEGs) using DEGseq (*q*-value < 0.001, |log$_2$FC| > 2) and edgeR (*q*-value < 0.05, |log$_2$FC| >2). The expression levels of genes were also measured by FPKM using Cufflinks 2.2.1 (http://cole-trapnell-lab.github.io/cufflinks/) for subsequent analysis. DEGs were analyzed using Cuffdiff [64] (*p*-value < 0.05, |log$_2$FC| >2).

### 4.7. Gene Expression Assay of HSPA1L Gene in Seven Tissues

We collected kidney, liver, mammary gland, myocardium, ovary, and spleen samples from another bull to compare *HSPA1L* gene expression between mature sperm and other tissues. The primers for *HSPA1L* and the housekeeping gene *GAPDH* were designed using the Primer-Blast tool, available on the NCBI website, and synthesized by Beijing Genomics Institute Tech (Beijing, China). (Table S1). Quantitative real-time PCR (qRT-PCR) analysis of each sample was performed in triplicate using SYBR™ green fluorescence (Roche, Penzberg, Germany), and the relative gene expression was normalized using *GAPDH* by the $2^{-\Delta\Delta Ct}$ method, as described previously [65].

### 4.8. Data Access

All high-throughput sequencing data were deposited in the NCBI GEO database under the accession number GSE131851.

## 5. Conclusions

In summary, using MZ twin seed dairy bulls with divergent reproductive traits, we were able to illustrate the conservation and variability of DNA methylation between individuals while controlling for genetic background, shared environmental exposure, age, gender, and cohort effects. Our findings improve our understanding of bovine sperm methylation patterns and provide a foundation for investigating the regulatory roles of DNA methylation on fertility traits of AI bulls.

**Supplementary Materials:** The following are available online at http://www.mdpi.com/2075-4655/3/4/21/s1.

**Author Contributions:** S.L. conducted the experiments, data analyses and manuscript writing. S.C. and W.C. performed pyrosequencing and bisulfite sequencing for the target regions. S.Z., S.C., Y.L. and H.Y. participated in semen and phenotypic data collection. A.L. and Y.W. provided the data of daughter fertility traits and calculated the estimated breeding values and their reliability. Y.Y. and G.E.L. participated in the result interpretation and paper revision. Y.Y. and S.Z. conceived and designed the study and revised the manuscript. All authors read and approved the final manuscript.

**Funding:** The work was supported by grants from the Modern Agro-Industry Technology Research System (CARS-36), the Beijing Natural Science Foundation (6182021), the National Natural Science Foundation of China (31272420), the Program for Changjiang Scholar and Innovation Research Team in University (IRT-15R62), the National Science and Technology Programs of China (2013AA102504, 2014ZX08009-053B) and Selection and Breeding for High Productive Dairy Cattle Program (NingXia [2016]3–6). The funders had no role in study design, data collection and analysis, decision to publish, or preparation of the manuscript.

**Acknowledgments:** We thank the Beijing Dairy Centre for providing data and semen samples of the monozygotic twin bulls.

**Conflicts of Interest:** The authors declare no competing interest.

## Abbreviations

| | |
|---|---|
| QTL | Quantitative trait locus |
| HMRs | Hypomethylated regions |
| DMR | Differentially methylated region |
| MBD-seq | Methyl-binding domain sequencing |
| MZ | Monozygotic |
| SNP | Single nucleotide polymorphism |
| WGBS | Whole genome-wide bisulfite sequencing |
| DMCs | Differentially methylated cytosines |
| GO | Gene ontology |
| RSAT | Regulatory Sequence Analysis Tools |
| FPKM | Fragments per kilobase of transcript per million mapped reads |
| DEGs | Differentially expressed genes |
| *HAPA1L* | Heat shock 70 kDa protein 1-like |
| GAPDH | Glyceraldehyde-3-phosphate dehydrogenase |
| TSS | Transcriptional start site |
| TTS | Transcriptional termination site |
| FDR | False discovery rate |
| O./E. | Observed/Expected |
| HOXA | Homobox A cluster |
| TF | Transcription factor |
| DMG | Differentially methylated genes |
| DEGs | Differentially expressed genes |
| *ACTN1* | Actinin alpha 1 |
| *ID1* | Inhibitor of DNA binding 1HLH protein |
| bHLH | Basic helix-loop-helix |
| *ZFYVE28* | Zinc finger FYVE-type containing 28 |
| EGFR | Epidermal growth factor receptor |
| EBVs | Estimated breeding values |

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
