# Peer review of "Divergence Analyses of Sperm DNA Methylomes between Monozygotic Twin AI Bulls"

_2075-4655, 2019_

Round 1

Reviewer 1 Report

The manuscript entitled "Divergences of sperm DNA methylomes between monozygotic twin seed bulls reveal fertility-related genes" is well designed work to assess regulatory mechanisms of seed bull sperm methylation and the roles of DNA methylation on offspring fertilities. Here author employed whole genome bisulfite sequencing to identify differentially methylated regions and genes in Bull semen.

However Authors need to check the manuscript for spelling errors. Also the discussion is very short and so suggest the author to elaborate on findings including the significance of this study in improving fertility in dairy bulls. More details on how differentially methylated genes influence fertility when bulls are older.

Author Response

Repsonse to reviewer 1:

Thank you for the opportunity to revise our manuscript on “Divergences of sperm DNA methylomes between monozygotic twin seed bulls reveal fertility-related genes”.

Comments:

However Authors need to check the manuscript for spelling errors. Also the discussion is very short and so suggest the author to elaborate on findings including the significance of this study in improving fertility in dairy bulls. More details on how differentially methylated genes influence fertility when bulls are older.

Response:  Thank you for your suggestion. We have checked and corrected the spelling errors. Based on your suggestion, we extended our discussion part. We discussed more about how sperm DNA methylation regulate bull fertility as well as embryo development (See L383-L404). We represented the potential effects of accumulated methylation differences when bulls are older on male fertility as well (See L377-L381).

Reviewer 2 Report

This is an interesting manuscript that in which the authors compared two monozygotic twin bulls who had different sperm parameters.  The authors performed an incredible amount of work, including confirming that the bulls were twins by sequencing 51,370 SNPs.  They performed whole genome bisulfite sequencing on the sperm of both bulls, and even provided extensive data on the fertility of some of the daughters of both bulls.  They identified significant differences in DNA methylation patterns.  This in an impressive study.

The only real criticism is that the authors should discuss how their work differs from that of Shojaei Saadi et al. (PMID:  26751019 – this reference should be updated to the final, published paper in 2017).  In that work, the authors performed a similar study on four pairs of bulls.  The authors should emphasize what is different between the two studies.

The English needs some work.  There are numerous grammatical errors.  This manuscript would benefit from an English editing service that would make it much more readable.

Author Response

Response to reviewer2:

Thank you for the opportunity to revise our manuscript on “Divergences of sperm DNA methylomes between monozygotic twin seed bulls reveal fertility-related genes”. 

Comments:

The only real criticism is that the authors should discuss how their work differs from that of Shojaei Saadi et al. (PMID: 26751019 – this reference should be updated to the final, published paper in 2017). In that work, the authors performed a similar study on four pairs of bulls.  The authors should emphasize what is different between the two studies.

Response: We apologize our mistake and thank you for pointing out the problem. We have updated the reference and discussed the differences between the two studies in L455-L470.

The English needs some work. There are numerous grammatical errors. This manuscript would benefit from an English editing service that would make it much more readable.

Response: Thank you for your suggestion. We have made essential revisions in the manuscript.

Round 2

Reviewer 2 Report

The authors have satisfied my concerns.